# Enhanced Interfacial Shear Strength and Critical Energy Release Rate in Single Glass Fiber-Crosslinked Polypropylene Model Microcomposites

**DOI:** 10.3390/ma11122552

**Published:** 2018-12-15

**Authors:** Uwe Gohs, Michael Thomas Mueller, Carsten Zschech, Serge Zhandarov

**Affiliations:** 1Leibniz-Institut für Polymerforschung Dresden e.V., Hohe Str. 6, D-01069 Dresden, Germany; mueller-michael@ipfdd.de (M.T.M.); zschech@ipfdd.de (C.Z.); serge.zhandarov@gmail.com (S.Z.); 2“V. A. Bely” Metal-Polymer Research Institute, National Academy of Sciences of Belarus, Kirov Str. 32a, 246050 Gomel, Belarus

**Keywords:** single fiber pull-out test, local interfacial shear strength, high energy electrons, crosslinked toughened polypropylene, glass fiber model microcomposites

## Abstract

Continuous glass fiber-reinforced polypropylene composites produced by using hybrid yarns show reduced fiber-to-matrix adhesion in comparison to their thermosetting counterparts. Their consolidation involves no curing, and the chemical reactions are limited to the glass fiber surface, the silane coupling agent, and the maleic anhydride-grafted polypropylene. This paper investigates the impact of electron beam crosslinkable toughened polypropylene, alkylene-functionalized single glass fibers, and electron-induced grafting and crosslinking on the local interfacial shear strength and critical energy release rate in single glass fiber polypropylene model microcomposites. A systematic comparison of non-, amino-, alkyl-, and alkylene-functionalized single fibers in virgin, crosslinkable toughened and electron beam crosslinked toughened polypropylene was done in order to study their influence on the local interfacial strength parameters. In comparison to amino-functionalized single glass fibers in polypropylene/maleic anhydride-grafted polypropylene, an enhanced local interfacial shear strength (+20%) and critical energy release rate (+80%) were observed for alkylene-functionalized single glass fibers in electron beam crosslinked toughened polypropylene.

## 1. Introduction

In the last years, the use of fiber-reinforced composites, mainly thermosets, in aerospace, aviation, building, wind energy, sport, leisure, and automotive uses has increased significantly [1]. Their continuous use in the automotive and transport industry is driven by the potential of reduced carbon dioxide emissions. However, high material and processing costs may influence its further growth in large-scale automotive production. Consequently, there is the need for alternative fiber-reinforced composites and fast-processing technologies. With respect to this requirement, textile thermoplastic composites offer a possible way for lightweight components. In Reference [2], highly integrated structures consisting of glass fiber (GF)–polypropylene (PP) hybrid yarn-based composites and an innovative technology demonstrator vehicle were developed. These hybrid yarns consisted of matrix and reinforcing filaments [3] and were used for the production of hollow structures by automated preforming technologies in order to fulfill the requirements of high-volume production [4]. The unconsolidated hollow structures provided good drapability and could be used for the production of complex shaped torsion- and bend-resistant structures.

Low fiber-to-matrix adhesion in the interphase of GF-reinforced PP mainly contributes to interfacial debonding and delamination, as well as limits its use. There are several approaches to enhancing the low interfacial adhesion between the GF and PP matrix. These are plasma treatment [5,6], the use of sizing [7], surface grafting [8,9], and the use of transcrystalline interphases [10]. In the case of continuous GF-reinforced PP composites, the simultaneous in situ commingling of glass and polypropylene filaments as well as the application of a water-based sizing containing aminosilane and maleic anhydride-grafted PP (MAH-g-PP) led to reduced fiber damage, a homogeneous mixing of both glass and PP filaments, good impregnation of glass filaments with the PP matrix, and the best performance of continuous GF-reinforced PP composites [11]. Besides aminosilane and MAH-g-PP, water-based sizing consists of additional low molecular components to protect the GF and to provide the required processing properties. In Reference [7], commercially sized glass fibers were analyzed. The results confirmed bonded and physisorbed sizing components on the analyzed GF surface. Up to 25 wt % of the sizing was not removable and was considered to be strongly chemically bonded to the GF surface. In addition, it has to be taken into account that reactive molecules must diffuse to the fiber surface during solution-based fiber modifications [12]. Consequently, degradation of reactive species generated in the bulk solution can occur. During the curing of fiber-reinforced thermosets, all sizing components can react with the functional groups of thermosetting resin [13], leading to physical and chemical interactions as well as high adhesion in the interphase. In the case of fiber-reinforced thermoplastics, no curing is involved. Consequently, the chemical reactions are limited to the GF fiber surface (interface_1) as well as the silane coupling agent and the MAH-g-PP (interface_2). This limits the enhancement of interfacial strength parameters in comparison to thermosets. In accordance with References [13,14], additional covalent bonds can be generated by a free radical process or a metallocene-catalyzed in situ copolymerization onto the fiber.

High energy electrons are used in many applications for polymerization, crosslinking, degradation, and functionalization of polymer materials [15]. This unique and sustainable technology uses the spatial and temporal precise generation of polymer radicals without the use of any additional chemical agents. In the case of PP, electron beam (EB) treatment leads to degradation and reduced mechanical properties [16,17,18]. Therefore, it has to be modified in order to allow EB-induced formation of covalent bonds (e.g., graftlinks or crosslinks) [19,20]. With respect to simultaneous in situ commingling of glass and polypropylene filaments, the requirements of melt spinnability have to be taken into account during the modification of PP in order to enable the preparation of continuous GF-reinforced PP composites via GF–PP hybrid yarn. In Reference [21], crosslinkable toughened PP (tPP) was produced by electron-induced reactive processing (EIReP) and was comprehensively tested with respect to mechanical performance [22] and melt spinnability [23,24]. In Reference [25], tPP and amino-sized single GFs were used for the preparation of single GF-reinforced tPP model microcomposites. After their EB treatment, a slightly enhanced interfacial adhesion and critical energy release rate were observed by analyzing the force-displacement curves of the single fiber pull-out test.

In Reference [26], EB curing was successfully applied to enhance the mechanical properties of carbon fiber acrylate composites. Based on References [23,24,25,26], this study was aimed at the impact of melt-spinnable crosslinkable tPP, alkylene-functionalized single GFs, and EB-induced chemical changes of interface_2 and matrix on the local interfacial shear strength (IFSS) and critical energy release rate (*G_ic_*) in single GF–PP model microcomposites. Altogether, single GF-reinforced PP model microcomposites were prepared and tested using four types of GF surface (non-, amino-, alkyl-, and alkylene-functionalized GF) and three types of PP (noncrosslinked PP, crosslinkable tPP, and crosslinked tPP) in order to systematically study the influence of GF functionalization (interface_1), adhesion promoter-to-matrix coupling (interface_2), and matrix crosslinking (interphase), separately. The basic idea of EB-induced chemical couplings is illustrated in Figure 1.

## 2. Experiment Section

### 2.1. Experiment

#### 2.1.1. Materials and Specimen Preparation

In accordance with References [23,24,25], we used PP HG455FB (93.5 ma %), ethylene octene copolymer (Engage 8100, 2.5 ma %), maleic anhydride-grafted PP (MAH-g-PP: Exxelor PO1020, 2 ma %), and a crosslinking agent (trimethylolpropane triacrylate (TMPTA), 2.0 ma %) in order to prepare melt-spinnable and crosslinkable tPP by EIReP. It was used as matrix material for the preparation of single E-glass fiber–PP model microcomposites for the pull-out test. The use of crosslinkable tPP enables the chemical modification of tPP (grafting, crosslinking) as well as the formation of chemical couplings between tPP matrix and alkylene-functionalized GF by EB treatment. Nonfunctionalized GF with an average diameter of 17 μm was melt-spun in accordance with Reference [11]. Any existing contamination on the GF surface was removed by a washing procedure using ethanol and acetone. Finally, the cleaned GF was modified by a UV treatment in ozone atmosphere (1 h). Afterwards, the cleaned fibers were immersed in 100 mL toluene (dry) with 1 mL of the corresponding adhesion promoter (10-undecenyltrimethoxysilane for EB, n-decyltrimethoxysilane for nonreactive coupling with tPP matrix) and heated at a temperature of 110 °C under reflux for 6 h. The coated E-glass fibers were dried for 2 h at a temperature of 130 °C under vacuum to safely prepare interface_1 (between GF and adhesion promoter) of our future GF–PP model microcomposites. Finally, the non-attached adhesion promoter molecules were rinsed off with ethanol. For comparison to the standard sizing procedure, cleaned fibers were also modified by applying an aqueous sizing consisting of 1 ma % coupling agent (3-aminopropyl-triethoxysilane, Table 1), 10 ma % film former (Aquacer 598, Table 1), and 89 ma % water by dip coating, followed by 6 h drying at 120 °C in an oven. All raw materials used and selected information are summarized in Table 1.

Toughened PP was prepared by continuous EIReP of a PP/ethylene octene copolymer (EOC) blend with 97.5/2.5 mass ratio at a dose of 6 kGy [21]. Afterwards, crosslinkable tPP was produced by nonreactive compounding of tPP (96 ma %), MAH-g-PP (Exxelor PO1020, 2 ma %), and a crosslinking agent (TMPTA, 2.0 ma %). Finally, nonfunctionalized, alkylene- (adhesion promoter-2), and alkyl- (adhesion promoter-3) functionalized single E-glass fibers were embedded in crosslinkable tPP matrix at a temperature of 180 °C under argon atmosphere. Two minutes after switching on the heating device, PP started to melt, and the fiber was embedded into the polymer matrix. After 5 min, the heating was switched off, and the temperature decreased with a cooling rate of about 5 K/min in order to reach ~25 °C. The total preparation time of the GF–PP model microcomposites amounted to 35 min. Afterwards, the pull-out test was started. In addition, nonfunctionalized and amino-functionalized (adhesion promoter-1) E-glass fibers were embedded in PP HG455FB (98 ma %)/MAH-g-PP (2 ma %) blend under the same conditions. Partially, GF–tPP model microcomposites were irradiated with a dose of 9 kGy in nitrogen atmosphere. The electron energy amounted to 1.5 MeV in order to ensure a homogeneous energy absorption in the model microcomposites. Based on the results of Reference [21], this dose was precisely selected in order to generate additional covalent bonds between alkylene-functionalized GF and the tPP matrix (interface_2), as well as within the tPP matrix (branching, crosslinking), for a systematic study of the influence of GF-to-matrix couplings and crosslinking of matrix on the interfacial strength parameters.

#### 2.1.2. Pull-Out Testing

All specimens were loaded into a specially designed pull-out apparatus constructed at the Leibniz Institute for Polymer Research Dresden e.V. [27]. All pull-out tests were carried out at room temperature. The initial pull-out rate amounted to 0.01 μm/s. After passing the maximum peak during the stage of initial loading, the pull-out rate was enhanced to 1 μm/s in order to reduce the time of measurement (~1.5 h). For each type of GF–PP model microcomposite (Table 3), 20 specimens were tested. Detailed information on the experimental procedure, the data acquisition, and the data processing in Mathematica were described in Reference [28]. Based on the experimental force-displacement curves, two characteristic force values (maximum force reached in the test, *F*_max_, and post-debonding force, *F_b_*) were determined and used for the calculation of the local interfacial shear strength (*τ_d_*), the interfacial frictional stress (*τ_f_*), and the critical energy release rate (*G_ic_*). These local interfacial strength parameters were determined by the “alternative” method [28] based on the maximum force reached in the test, *F*_max_, and the post-debonding force, *F_b_*.

#### 2.1.3. Sample Designation and Composition of Single GF–PP Model Microcomposites

In this study, six single GF-reinforced PP model microcomposites were prepared using four types of GF surface (non-, amino-, alkyl-, and alkylene-functionalized GF) and three types of PP (non crosslinked PP, crosslinkable tPP, and crosslinked tPP). In Table 2, some abbreviations are introduced in order to enhance readability. In accordance with the dose absorbed during the EB treatment, the matrices of model microcomposites are designated by the subscripts “non” and “cross” for noncrosslinked and crosslinked matrices, respectively. The crosslinking was achieved by an EB treatment with a dose of 9 kGy. The dose was defined as absorbed energy per unit of mass (unit: kGy), and controlled the number of free radicals generated per polymer chain. The sample designation and the composition of GF–PP model microcomposites are summarized in Table 3.

The use of PP_non_ and amino-sized GF (GF_amino_) represented state of the art technology for the preparation of GF-reinforced PP [11] and was the reference model microcomposite system of this study. Since PP is highly nonpolar, the presence of polar functional groups in MAH-g-PP enhanced compatibility with polar GF [29], as well as led to a homogenous dispersion of the crosslinking agent TMPTA (2 ma %) and a high crosslinking efficiency of tPP [21].

#### 2.1.4. Scanning Electron Microscopy (SEM)

The fracture surface of single glass fibers after the pull-out test was investigated by a LEO 435 VP Ultra plus Scanning Electron Microscope (SEM) from Carl Zeiss SMT, Oberkochen, Germany, in order to get the first information on the morphological structure of the break area. A secondary electron detector was used to produce a topographic SEM image. All single glass fibers were sputter-coated with 3 nm platinum prior to the SEM analysis.

## 3. Results and Discussion

### 3.1. Force-Displacement Curves

In Figure 2, representative force-displacement curves are shown for each type of single GF–PP model microcomposite prepared. These force-displacement curves clearly demonstrate the problems that arose during the determination of the kink where the force was able to initiate the interfacial debonding. This point of the force-displacement curve corresponds to the debond force (*F_d_*) and is used in the traditional method for the estimation of the local interfacial shear strength. However, this kink point was hardly discernible for some specimens (see inserts in Figure 2). In contrast to *F_d_*, the maximum value of force reached in the single fiber pull-out test (*F*_max_) and the post-debond force (*F_b_*) were well observable and could be determined with higher accuracy in comparison to the debond force. In further consideration of Figure 2a–d, a parasite peak was visible during the stage of initial loading. This peak was observed after passing the maximum peak and was related to the change of the pull-out rate during the measurement (see Section 2.1.2.). Consequently, the first local maximum of the force-displacement curve was taken in order to determine *F*_max_ for these model microcomposites.

In conclusion, these experimental force displacement curves impressively confirmed that the “alternative” method was the method of choice for the analysis of experimental force-displacement curves. It is the most accurate and reliable analyzing method. In contrast, the traditional method often yields in local interfacial strength parameters with enlarged uncertainty due to slight changing of the slope of the force-displacement curve, no kink or multiple kinks. Multiple kinks can be artifacts resulting from the noncylindrical shape of model composites or can be related to the crack initiation in the glue that holds the opposite end of the fiber. Consequently, the wrong kink can be erroneously taken for the determination of debond force.

### 3.2. Evaluation of Pull-Out Test

The mean values of fiber diameter (*d_f_*), embedding length (*l_e_*), *F_max_*, *F_b_*, and the interfacial strength parameters (*τ_d_*, *τ_f_*, *G_ic_*) are summarized in Table 4 for all types of single GF–PP model microcomposites. All experimental uncertainties of mean values were related to an assurance level of 68%. As expected, the nonfunctionalized polar glass filament embedded in the PP/MAH-g-PP blend (MC1) showed the lowest interfacial parameters. Within the experimental uncertainty, these values were comparable with those of alkyl-functionalized (adhesion promoter-3) nonpolar single GF embedded in crosslinked tPP (MC4). Consequently, crosslinking of tPP matrix in the absence of functionalized GF-to-matrix coupling (interface_2) had no influence on the interfacial strength parameters. In the case of nonfunctionalized polar single GF in crosslinked tPP model microcomposites (MC3), slightly enhanced interfacial parameters were observed, which might be explained by the additional interaction of polar TMPTA with the nonfunctionalized polar GF surface. As expected, amino-functionalized GF showed a significant enhancement of *τ_d_* and *G_ic_*. These results were in good agreement with Reference [11]. Unexpectedly, the same level of interfacial properties was observed for alkylene-functionalized single GF in noncrosslinked tPP. In accordance with Figure 1, EB treatment was required in order to generate covalent bonds between alkylene-functionalized single GF and tPP (interface_2). Thus, we conclude that during the preparation of glass filament tPP model composites, temperature-initiated radical grafting reactions occurred due to the extended handling of tPP at a temperature of 180 °C (see Section 2.1.1.). The maximum local interfacial shear strength (+20%) and critical energy release rate (+80%) were observed for alkylene-functionalized single GF in crosslinked tPP in comparison to amino-functionalized single GF in the PP/MAH-g-PP blend. This result confirmed our approach for EB-induced enhancement of interfacial strength parameters in GF–tPP model microcomposites by the formation of alkylene-functionalized GF-to-matrix covalent bonds and matrix crosslinking (interface_3). Consequently, three different interfaces had to be designed in order to enhance the interfacial strength parameters.

In addition, it has to be noted that the interfacial frictional stress (Figure 3c) only slightly depended on the type of GF functionalization and the composition of PP matrix used. Finally, the calculated interfacial strength parameters are shown in Figure 3 for a better visualization. As can be easily seen, the results of *τ_d_* and *G_ic_* demonstrated that the energy-based parameter *G_ic_* was more sensitive to chemical changes in the GF–PP interphase in comparison to the stress-based parameter *τ_d_*. This is not surprising, since, as was shown in Reference [30], the relationship between *G_ic_* and *τ_d_* is a quadratic equation:(1)$$G_{ic}=\left[c_{0}{\left(\Delta T\right)}^{2}+c_{1}\Delta T\tau_{d}+c_{2}\tau_{d}^{2}\right]d_{f},$$
where ∆*T* = *T_test_* − *T_ref_* is the difference between the test temperature and the reference stress-free temperature; *c*_0_, *c*_1_, and *c*_2_ are functions of material constants but not of specimen geometry; and *d_f_* is the diameter of fiber. For polymer matrices, *T_ref_* is considered to be equal to the glass transition temperature (*T_g_*) if *T_g_* > *T_test_*, but *T_ref_* = *T_test_* (i.e., ∆*T* = 0) if *T_g_* < *T_test_* [31]. In the case of PP, the glass transition temperature amounted to −10 °C, which was lower than the test temperature during the pull-out test (25 °C). Consequently, ∆*T* = 0, and *G_ic_* is proportional to *τ*^2^*_d_*. In other words, the local IFSS is proportional to the debond force *F_d_*, but the critical energy release rate is proportional to the square of the debond force and is therefore much more sensitive. In the case of *T_g_* > *T_test_*, the relationship between *τ_d_* and *G_ic_* is more complicated for these polymer matrices.

### 3.3. Evaluation of GF Surface after the Pull-Out Test

Figure 4 shows SEM micrographs of the GF surface after the pull-out test. In the case of MC1 (Figure 4a,b), a very smooth GF surface with some small zones of attached polymer and the typical morphology of a brittle fracture behavior were observed. This indicated a low GF–PP adhesion and was in agreement with the lowest values of local interfacial shear strength and critical energy release rate. On the other hand, more zones of attached polymer were observed for MC3 (Figure 4e), MC4 (Figure 4g), and MC5 (Figure 4i). In the case of MC2 (Figure 4c) and MC6 (Figure 4k), large zones of polymer were observed, indicating an enhanced GF-matrix adhesion and confirming the maximum values of local interfacial shear strength and critical energy release rate. With respect to an enhanced GF-matrix adhesion, a polymer layer on the glass fiber surface was required. At higher magnification (right figures), different morphologies of polymer were observed at the GF surface. In the case of MC1 (Figure 4b), some polymer was observed at the GF surface. On the other hand, the GF of MC3 (Figure 4f) and MC4 (Figure 4h) showed a partial polymer layer at their surface. Nevertheless, these coverages were not homogeneous. A homogeneous and rough layer was observed at the GF surface of MC2 (Figure 4d), MC5 (Figure 4j), and MC6 (Figure 4l). This was in agreement with the experimental results of *G_ic_* and indicated an enhanced toughness. As shown in Figure 4c,k, the PP matrix partially adhered to the GF surface, and the fracture in the GF model microcomposite occurred in the polymer matrix surrounding the GF. In the case of MC6 (Figure 4l), the fracture surfaces showed shear yielding structures that were formed by shear yielding during the pull-out test.

Further knowledge on the adhesion and the fracture mechanism could be gained from the SEM micrographs of the pull-out holes (Figure 5). The morphological analysis showed significant differences between MC1, MC2, MC5, and MC6. In the case of MC1 (Figure 5a), the edge layer of the pull-out hole showed no deformation and confirmed a low GF-matrix adhesion and a brittle fracture behavior. In contrast, the pull-out holes of MC2, MC5, and MC6 showed a strongly plastically deformed edge layer, indicating higher GF-matrix adhesion and enhanced ductility and toughness in comparison to the nonfunctionalized GF (MC1). These experimental results were in good agreement with the experimental data of *G_ic_*.

## 4. Conclusions

The authors compared different types of modifications of GF surface and PP matrix in order to estimate the local interfacial strength parameters from force-displacement curves recorded in single fiber pull-out tests. The experimental force-displacement curves were analyzed by the robust “alternative” method based on the maximum force (*F*_max_) and post-debonding force (*F_b_)*. The results of single fiber pull-out tests highlighted the chemical modification of the GF–PP interphase and PP matrix by EB treatment in order to enhance the local interfacial shear strength and the critical energy release rate of the interphase for single GF-reinforced crosslinked tPP model microcomposites. The crosslinking of tPP matrix in the absence of functionalized GF-to-matrix coupling (interface_2) had no influence on the interfacial strength parameters. To our present knowledge, three interfaces had to be tailored in order to get enhanced local interfacial strength parameters. These were the functionalized GF surface–adhesion promoter interface (interface_1), the adhesion promoter–PP interface (interface_2), and the adhesion promoter grafted PP–crosslinked PP interface (interface_3/interphase_3). In the case of single GF–crosslinked tPP model microcomposites, the IFSS amounted to (32.8 ± 1.5) MPa. Within experimental uncertainty, this value was comparable to the tensile strength of crosslinked tPP, (31.9 ± 0.7) MPa [25]. First, information on the adhesion and fracture behavior was gained from SEM micrographs of the fiber surface after the pull-out test and the pull-out holes. In the absence of GF surface and matrix modification (MC1), a very smooth GF surface with the typical morphology of a brittle fracture behavior was observed. In contrast, the fracture occurred in the polymer matrix surrounding the GF, and the rough fracture surfaces showed shear yielding structures after GF surface modification as well as crosslinking of matrix and interphase (MC6). This ductile fracture behavior led to the largest critical energy release rate.

Further investigations are required in order to understand the role of physical and chemical interactions within the interphase of GF-reinforced thermoplastics. In addition, comprehensive studies are in preparation in order to study the influence of the crosslinking degree of tPP matrix on the local interfacial strength parameters at differently designed interface_1 and interface_2 in order to prove the possibility of preparing continuous GF-reinforced recyclable (long-chain branched, but noncrosslinked) toughened PP composites with enhanced local interfacial strength parameters.

## Figures and Tables

**Figure 1 materials-11-02552-f001:**
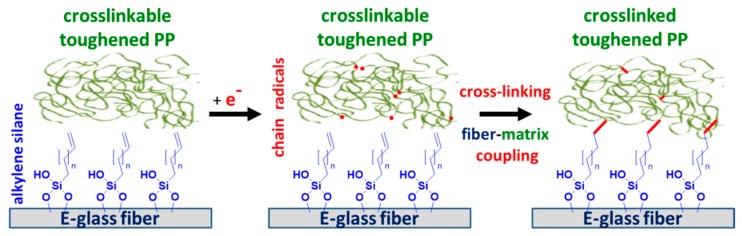
Schematic of electron beam (EB)-induced glass fiber (GF)–toughened polypropylene (tPP) couplings in the interface and crosslinking of tPP.

**Figure 2 materials-11-02552-f002:**
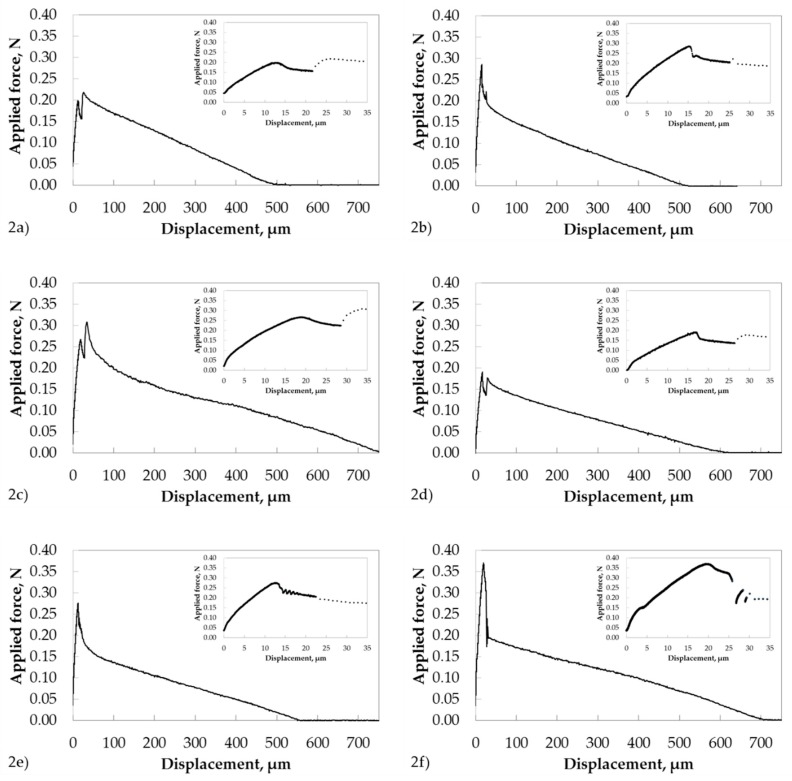
Representative force-displacement curves of single GF–PP model microcomposites: (**a**) MC1; (**b**) MC2; (**c**) MC3; (**d**) MC4; (**e**) MC5; and (**f**) MC6.

**Figure 3 materials-11-02552-f003:**
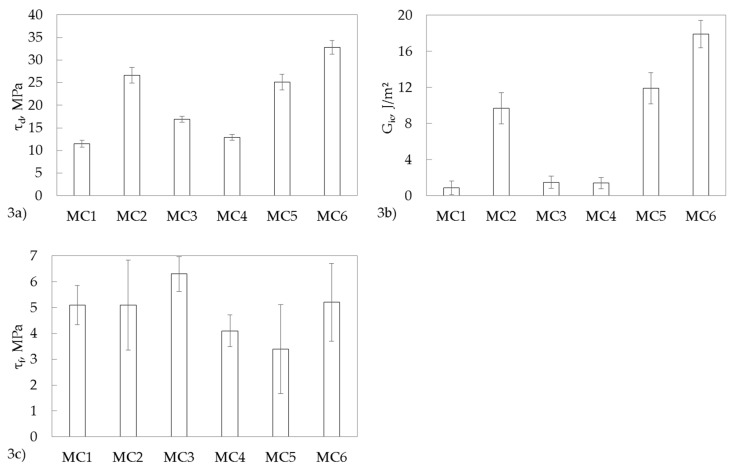
(**a**) Local interfacial shear strength; (**b**) critical energy release rate; and (**c**) interfacial friction strength of single GF–PP model microcomposites.

**Figure 4 materials-11-02552-f004:**
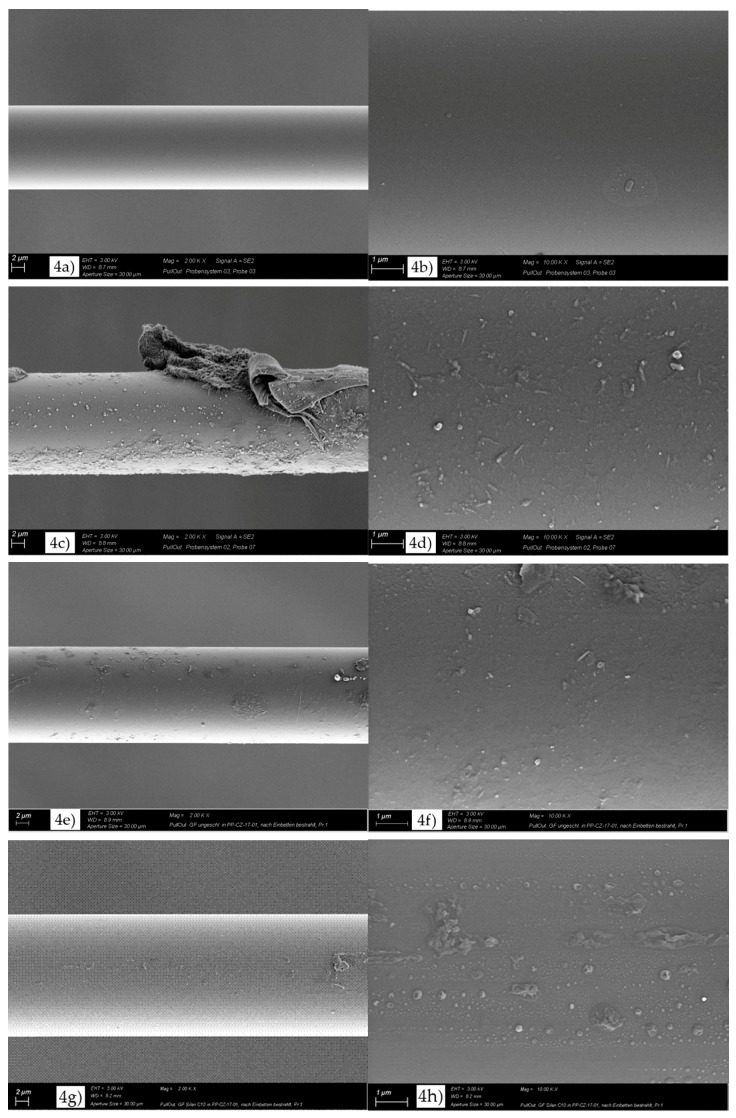
SEM micrographs of GF surfaces for (**a,b**) MC1; (**c,d**) MC2; (**e,f**) MC3; (**g,h**) MC4; (**i,j**) MC5; and (**k,l**) MC6 after the pull-out test.

**Figure 5 materials-11-02552-f005:**
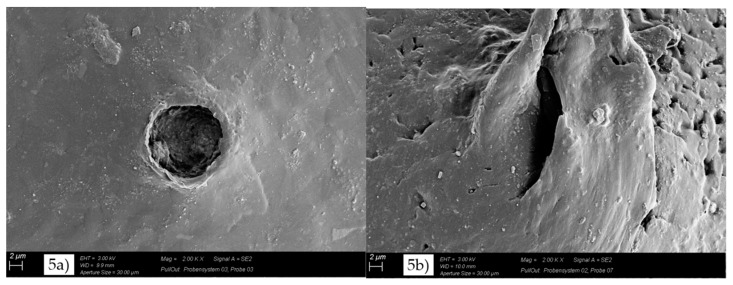
SEM micrographs of pull-out hole for (**a**) MC1; (**b**) MC2; (**c**) MC5; and (**d**) MC6.

**Table 1 materials-11-02552-t001:** Raw materials and selected information.

Raw Material	Type	Supplier	Additional Information
polypropylene	PP HG455FB	Borealis AG, Linz, Austria	melt flow rate: 27 g/10 min (230 °C/2.16 kg)
ethylene octene copolymer	Engage 8100	Dow Chemical Company, Midland, MI, USA	melt flow rate: 1.0 g/10 min (190 °C/2.16 kg)
maleic anhydride-grafted PP	Exxelor PO1020	Exxon Mobil Corporation, Antwerp, Belgium	0.5 to 1 ma % grafted maleic anhydride
trimethylol-propane triacrylate	Trimethylolpropane triacrylate (TMPTA)	Cytec Surface Specialities, Vlaardingen, The Netherlands	Chemical Abstracts Service (CAS): 15625-89-5trifunctional
PP-film former	Aquacer 598	BYK-Chemie GmbH, Wesel, Germany	0.25 to 0.5 ma % grafted maleic anhydride
adhesion promoter-1	Dynasylan AMEO, 3-aminopropyl-triethoxysilane	Evonik Industries, Marl, Germany	CAS: 919-30-2 bifunctional
adhesion promoter-2	10-undecenyltrimethoxysilane	Gelest, Inc., Morrisville, NC, USA	CAS: 872575-06-9 bifunctional, for EB
adhesion promoter-3	n-decyltrimethoxysilane	Gelest, Inc., Morrisville, NC, USA	CAS: 5575-48-4 monofunctional

**Table 2 materials-11-02552-t002:** Abbreviations for functionalized GF surfaces and matrices used.

Abbreviation	Composition
PP_non_	blend of PP HG455FB (98 ma %) and Exxelor PO1020 (2 ma %), noncrosslinked
tPP_non_	blend of PP HG455FB (94 ma %), Engage 8100 (2 ma %), TMPTA (2 ma %), and Exxelor PO1020 (2 ma %), noncrosslinked
tPP_cross_	blend of PP HG455FB (94 ma %), Engage 8100 (2 ma %), TMPTA (2 ma %), and Exxelor PO1020 (2 ma %), crosslinked
GF_non_	nonfunctionalized GF, but pretreated (see Section 2.1.1.)
GF_amino_	GF sized with aqueous standard sizing consisting of a coupling agent (3-aminopropyl-triethoxysilane) and a film former (Aquacer 598)
GF_alkyl_	alkyl-functionalized GF
GF_alkylene_	alkylene-functionalized GF + 0 kGy

**Table 3 materials-11-02552-t003:** Sample designation and composition of GF–PP model microcomposites.

Sample Designation	Composition
MC1	PP_non_ + GF_non_
MC2	PP_non_ + GF_amino_
MC3	tPP_cross_ + GF_non_
MC4	tPP_cross_ + GF_alkyl_
MC5	tPP_non_ + GF_alkylene_
MC6	tPP_cross_ + GF_alkylene_

**Table 4 materials-11-02552-t004:** Experimental and interfacial strength parameters for single glass fiber polypropylene model microcomposites.

Type	*d_f_*, μm	*l_e_*, μm	*F*_max_, N	*F_b_*, N	*τ_d_*, MPa	*τ_f_*, MPa	*G_ic_*, *J*/m²
MC1	17.2 ± 0.3	592 ± 25	0.195 ± 0.009	0.163 ± 0.010	11.5 ± 0.8	5.5 ± 0.3	0.9 ± 0.4
MC2	16.6 ± 0.2	615 ± 15	0.288 ± 0.098	0.163 ± 0.019	26.6 ± 1.7	5.1 ± 0.6	9.7 ± 1.3
MC3	16.0 ± 0.4	633 ± 15	0.263 ± 0.014	0.216 ± 0.013	16.9 ± 0.7	6.3 ± 0.4	1.5 ± 0.2
MC4	17.0 ± 0.8	680 ± 14	0.194 ± 0.014	0.146 ± 0.013	12.9 ± 0.6	4.1 ± 0.3	1.4 ± 0.3
MC5	17.3 ± 0.4	655 ± 25	0.267 ± 0.023	0.122 ± 0.010	25.1 ± 1.7	3.4 ± 0.3	11.9 ± 1.9
MC6	16.9 ± 0.3	620 ± 32	0.351 ± 0.029	0.166 ± 0.013	32.8 ± 1.5	5.2 ± 0.2	17.9 ± 2.5

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
