# Peer review of "Enhanced Interfacial Shear Strength and Critical Energy Release Rate in Single Glass Fiber-Crosslinked Polypropylene Model Microcomposites"

_materials, 2018, doi:10.3390/ma11122552_

Round 1
Reviewer 1 Report
The paper presents an experimental investigation of the shear strength and critical energy release rate in single glass fiber/polypropylene model micro-composites. Both the role of surface treatment of the glass fibers and the role of cross linking the PP matrix and possibly developing crosslinks between the PP matrix and the PP chains chemically attached to the interface are investigated by comparing specifically designed assemblies. As can be expected, the largest critical energy release rate is obtained for fully cross linked (bulk and interface) system, while the interfacial shear strength appears to be quite insensitive to the surface treatment and to cross linking. The paper is clearly written, and the results will be useful, even if there is no real analysis or discussion of the underlying mechanisms. I recommend acceptance, in the present form of the paper.
Author Response
Reviewer 1:
The paper presents an experimental investigation of the shear strength and critical energy release rate in single glass fiber/polypropylene model micro-composites. Both the role of surface treatment of the glass fibers and the role of cross linking the PP matrix and possibly developing crosslinks between the PP matrix and the PP chains chemically attached to the interface are investigated by comparing specifically designed assemblies. As can be expected, the largest critical energy release rate is obtained for fully cross linked (bulk and interface) system, while the interfacial shear strength appears to be quite insensitive to the surface treatment and to cross linking. The paper is clearly written, and the results will be useful, …. I recommend acceptance, in the present form of the paper.
Response: We are delighted to learn that the honorable reviewer has appreciated our work. We would like to thank the reviewer for providing valuable suggestion. Accordingly, we have addressed the comment of the reviewer and modified the manuscript.
1) The paper is clearly written, and the results will be useful, even if there is no real analysis or discussion of the underlying mechanisms.
Response: We added the subsection 3.3. “Evaluation of GF surface after pull-out test” in order to show SEM micrographs of the GF surface/fracture surface after the pull-out test and to discuss differences in the morphological structure.
Reviewer 2 Report
This study by Gohs et. al. is certainly interesting and relevant. I have two primary comments on the study.
1) The structure of the material systems is not investigated. While the interfacial shear strength is characterized and analyzed in some detail, the morphological structure is completely ignored. It would be helpful to get some idea of the fracture surface, via electron or optical microscopy.
2) On similar lines, could the authors add some more details of the physical characterization of the polymer/composite system. Some data in the form of IR spectra could be helpful in truly understanding the interfacial structure.
Additionally, the conclusions of the present are too weak and not properly drafted. I would suggest rewriting the conclusions to comment on the stated objective of the work, instead of commenting on the conclusions reported in literature.
Author Response
Reviewer 2:
I have two primary comments on the study.
Response: The Authors would like to thank the reviewer for the evaluation of our manuscript. The reviewer’s comments were very helpful in improving the manuscript. The Authors have addressed the comments raised by the reviewer.
1) The structure of the material systems is not investigated. While the interfacial shear strength is characterized and analyzed in some detail, the morphological structure is completely ignored. It would be helpful to get some idea of the fracture surface, via electron or optical microscopy.
Response: We added the subsection 3.3. “Evaluation of GF surface after pull-out test” in order to show SEM micrographs of the GF surface/fracture surface after the pull-out test and to discuss differences in the morphological structure.
3.3. Evaluation of GF surface after the pull-out test
Figure 4 shows SEM micrographs of the GF surface after the pull-out test. In the case of MC1 (Figure 4a, 4b), a very smooth GF surface with some small zones of attached polymer and the typical morphology of a brittle fracture behavior was observed. This indicated a low GF/PP adhesion and was in agreement with the lowest values of local interfacial shear strength and critical energy release rate. On the other hand, more zones of attached polymer were observed for MC3, MC4 and MC5. In the case of MC2 and MC6, large zones of polymer were observed indicating an enhanced GF/matrix adhesion and confirmed the maximum values of local interfacial shear strength and critical energy release rate. With respect to an enhanced GF/matrix adhesion, a polymer layer on the fiber surface is required. At higher magnification (right figures), different morphologies of polymer were observed at the GF surface. In the case of MC1, some polymer was observed at the GF surface. On the other hand, the GF of MC3 and MC4 showed a partial polymer layer at their surface. Nevertheless, these coverages are not homogeneous. A homogeneous and rough layer was observed at the GF surface of MC2, MC5 and MC6. This is in agreement with the experimental results of Gic and indicated an enhanced toughness. As shown in Figure 4c and 4k, the PP matrix partially adhered to the GF surface and the fracture in the GF model microcomposite occurred in the polymer matrix surrounding the fiber. In the case of MC6 (Fig. 4l), the fracture surfaces showed shear yielding structures which were formed by shear yielding during the pullout test.
Figure 4. SEM micrographs GF-PP model composites after pull-out test (MC1: Fig. 4a,b; MC2: Fig. 4c,d; MC3: Fig. 4e,f; MC4: Fig. 4g,h; MC5: Fig. 4i,j; MC6: Fig. 4k,l).
Further knowledge on the adhesion and the fracture mechanism can be gained from the SEM micrographs of the pullout holes (Fig. 5). The morphological analyze showed significant differences between MC1, MC2, MC5 and MC6. In the case of MC1 (Fig. 5a), the edge layer of the pullout hole showed no deformation and confirmed a low GF/matrix adhesion and a brittle fracture behavior. In contrast, the pullout holes of MC2, MC5 and MC6 showed strongly plastically deformed edge layer indicating higher GF/matrix adhesion and enhanced ductility/toughness in comparison to unsized GF (MC1). These experimental results are in good agreement with the experimental data of Gic.
Figure 5. SEM micrographs of pull-out hole for MC1 (Fig. 5a), MC2 (Fig. 5b), MC5 (Fig. 5c) and MC6 (Fig. 5d).
2) On similar lines, could the authors add some more details of the physical characterization of the polymer/composite system. Some data in the form of IR spectra could be helpful in truly understanding the interfacial structure.
Response: We would like to thank the reviewer for this valuable suggestion. This work was designed to investigate the new approach for fiber surface treatment as well as GF/matrix interphase and matrix modification by electron beam treatment. Based on the experimental results further investigations are running in order to understand the role of physical and chemical interactions within the interphase of GF reinforced thermoplastics. SEM of single E-GF after the modification of GF surface (before the preparation of model microcomposite) and after pullout test as well as their corresponding energy dispersive X-ray (EDX) spectrum will be used for morphological analysis. The EDX spectrum of non-sized single E-GF (peaks of Al, Ca O, Si) will be used as standard. In the case of differently sized E-glass fibers, the “C” peak will be used for the evaluation of the GF surface before the preparation model microcomposites and GF/matrix interphase (C/Si ratio) after pullout test. In addition, differential scanning calorimetry and thermogravimetrical analysis will be used for the evaluation of GF-surface modification and GF/matrix interphase after the pull-out test.
3) Additionally, the conclusions of the present are too weak and not properly drafted. I would suggest rewriting the conclusions to comment on the stated objective of the work, instead of commenting on the conclusions reported in literature.
Response: We changed the conclusion in accordance to your comments. In absence of GF surface and matrix modification (MC1), a very smooth GF surface with the typical morphology of a brittle fracture behavior was observed by SEM after the pullout test. In contrast, the fracture occurred in the polymer matrix surrounding the GF and the fracture surfaces showed shear yielding structures after GF surface modification as well as crosslinking of matrix and interphase (MC6). This ductile fracture behavior led to the largest critical energy release rate.
Reviewer 3 Report
Interesting, correctly edited article important from the practical point of view. A large number of tests are efficiently presented in a condensed but clear way. Test results well documented and analyzed. Conclusions are presented in a descriptive way. They are drawn on the basis of conducted research and based on their results and supported and documented by literature overview. The literary is quite extensive, well matched to the needs of the article. The final evaluation of the article is very good.
Author Response
Reviewer 3:
Interesting, correctly edited article important from the practical point of view. A large number of tests are efficiently presented in a condensed but clear way. Test results well documented and analyzed. Conclusions are presented in a descriptive way. They are drawn on the basis of conducted research and based on their results and supported and documented by literature overview. The literary is quite extensive, well matched to the needs of the article. The final evaluation of the article is very good.
Response: We would like to thank the reviewer for the evaluation of our work.
Reviewer 4 Report
Reviewed manuscript describes research evaluation titled “Enhanced interfacial shear strength and critical energy in a glass fiber-crosslinked polypropylene model microcomposites”. Composite and microcomposite materials are used all over the world and are used in industry, transportation, construction, infrastructure, construction, sports and a commercial market. Also, the use of composites is a necessity related to the reduction of energy use and CO2 emissions.
The work contains a proper review of 30 literature articles, including 8 papers of the authors of this manuscript. The tests on modificated glass fiber reinforced polypropylene composites were carried out properly. Three interfaces were selected in order to get enhanced local interfacial strength parameters (the functionalized GF surface – adhesion promoter interface, the adhesion promoter - PP interface, the adhesion promoter grafted PP – crosslinked PP interface). The authors should explain in detail why PP grafted with maleic anhydride was used for the study. There is a strong need in the future for additional studies aimed at exploring which microcomposities are most effective.
Author Response
Reviewer 4:
Reviewed manuscript describes research evaluation titled “Enhanced interfacial shear strength and critical energy in a glass fiber-crosslinked polypropylene model microcomposites”. Composite and microcomposite materials are used all over the world and are used in industry, transportation, construction, infrastructure, construction, sports and a commercial market. Also, the use of composites is a necessity related to the reduction of energy use and CO2 emissions.
The work contains a proper review of 30 literature articles, including 8 papers of the authors of this manuscript. The tests on modificated glass fiber reinforced polypropylene composites were carried out properly. Three interfaces were selected in order to get enhanced local interfacial strength parameters (the functionalized GF surface – adhesion promoter interface, the adhesion promoter - PP interface, the adhesion promoter grafted PP – crosslinked PP interface).
Response: We are delighted to learn that the honorable reviewer has appreciated our work. In addition, we would like to thank the reviewer for the valuable suggestion. We have taken into account the comment and modified the manuscript.
1) The authors should explain in detail why PP grafted with maleic anhydride was used for the study. There is a strong need in the future for additional studies aimed at exploring which microcomposities are most effective.
Response: We added additional information on the use of maleic anhydride grafted PP.
The combination of PPnon and amino sized GF (GFamino) is state of the art technology for the preparation of GF reinforced PP [11] and the reference model microcomposite system of this study. Since PP is highly non-polar, the presence of polar functional groups in MAH-g-PP enhances the compatibility with polar GF [29] as well as leads to a homogenous dispersion of the crosslinking agent TMPTA (2 ma.%) and a high crosslinking efficiency of tPP [21].
Round 2
Reviewer 2 Report
The authors have made serious efforts to improve the manuscript. I suggest publication i the current form.